# Development of Single Nucleotide Polymorphism and Association Analysis with Growth Traits for Black Porgy (*Acanthopagrus schlegelii*)

**DOI:** 10.3390/genes13111992

**Published:** 2022-10-31

**Authors:** Zhiwei Zhang, Zhijie Lin, Mingliang Wei, Ziqiang Chen, Mingjun Shen, Guangyong Cao, Yue Wang, Zhiyong Zhang, Dianchang Zhang

**Affiliations:** 1Jiangsu Marine Fishery Research Institute, Nantong 226007, China; 2National Demonstration Center for Experimental Fisheries Science Education, Shanghai Ocean University, Shanghai 201306, China; 3South China Sea Fishery Research Institute, Chinese Academy of Fishery Sciences, Guangzhou 510300, China

**Keywords:** black porgy, transcriptome, MALDI-TOF MS, SNPs, genetics and breeding

## Abstract

Black porgy is an important marine aquaculture fish species whose production is at the fifth position in all kinds of marine-cultured fishes in China. In this study, Illumina high-throughput sequencing technology was used to sequence the total RNA of black porgy. Sixty-one candidate SNPs (Single Nucleotide Polymorphism) were screened out and genotyped through GATK4 (Genome Analysis ToolKit) software and MALDI-TOF MS (Matrix-Assisted Laser Desorption/ Ionization Time of Flight Mass Spectrometry). The experimental results showed that a total of sixty SNPs were successfully genotyped, with a success rate of 98.36%. The results of principal component analysis and correlation analysis of growth traits showed that body weight was the first principal component, with a cumulative contribution rate of 74%. There were significant correlations (*p* < 0.05) or extremely significant correlations (*p* < 0.01) between different growth traits. The results of genetic parameter analysis and association analysis showed that scaffold12-12716321, scaffold13-4787950, scaffold2-13687576 and scaffold290-11890 were four SNPs that met the requirement of polymorphic information content and conformed to the Hardy–Weinberg equilibrium. There were significant differences between their genotype and the phenotype of growth traits. The four SNP molecular markers developed in this research will lay a foundation for further exploration of molecular markers related to the growth traits of black porgy and will provide a scientific reference for the further study of its growth mechanisms. At the same time, these molecular markers can be applied to the production practices of black porgy, so as to realize selective breeding at the molecular level and speed up the breeding process.

## 1. Introduction

Black porgy (*Acanthopagrus schlegelii*), belonging to *Perciformes*, *Sparidae*, which is a kind of nearshore and demersal fish with a wide tolerance of temperature and salinity, is mostly distributed in the coastal areas of China, Korea, Japan and Vietnam. Black porgy is one of the most popular and important commercial breeds in China and East Asian countries because of its high nutritional value, strong resistance to harsh environmental factors and fast growth [1,2,3]. However, in the northern part of the Yangtze River, the black porgy should be moved indoors during winter because of the low temperature outdoors. It takes three years for the fish to grow into commodity fish. This factor greatly limits development of its industrialization [2]. The total production of black porgy is about 60,000 tones per year in China [4], which is at the fifth position in all kinds of marine-cultured fishes, just behind the large yellow croak, sea bass, grouper and flounder. A complete industry chain for black porgy aquaculture has been formed, from its breeding, aquaculture, processing and sales, which plays an important role in the marine fish culture. In recent years, with the innovation and development of molecular biology technology, more and more new techniques have been applied to production practices. Molecular markers are important resources for the application of molecular biology and multi-omics in fish genetic breeding and aquaculture production activities. Screening out effective molecular markers through base sequence differences is useful, not only for the study of genetic diversity of fish, but also for effective breeding of improved varieties at the molecular genetic level, which will greatly promote the development and utilization of fish germplasm resources [5,6].

All the genetic information required for fish ontogeny is stored in nucleic acid sequences. Therefore, a lot of biological information that is related to individual growth and development are contained in the genome, transcriptome and proteome sequences. Transcriptome technology plays a non-negligible role in the research of aquatic organism genetics and breeding, growth and development, disease resistance and immunity, species evolution, etc., [7,8]. It is possible for us to comprehensively and quickly obtain almost all transcript sequence information in one state of specific tissues or organs of a species through next-generation high-throughput sequencing [9]. Single nucleotide polymorphisms (SNPs) are the most common, widely distributed and heritable variations in plant and animal genomes. As third-generation molecular genetic markers, SNPs were first proposed in 1996 and have gradually become one of the mainstream molecular markers. They have been widely used in the fields of germplasm identification, genetic diversity analysis and molecular genetic breeding [10,11], especially the markers from transcripts, because cDNA is an important part of expressed genes and is considered as the most promising new generation of molecular markers [12]. With the rapid development of high-throughput sequencing technology and the continuous reduction of experimental costs, genotyping technologies such as sequencing and time-of-flight mass spectrometry have allowed many important achievements in the field of fish molecular marker development [13]. Two SNPs markers that have significant correlation with growth traits were identified in the IGF1 and IGF1R genes of *Pangasianodon hypophthalmus* by using the multiplex SNaPshot method [14]. SNPs have been widely used for fish growth traits *Siniperca chuatsi* [10], *Paramisgurnus dabryanus* [15] and *Micropterus salmoides* [16], included in sex determination, such as in *Ictalurus punctatus* [17], genetic diversity in *Mylopharyngodon piceus* [18], *Schizothorax wangchiachii* [19] and *Anguilla japonica* [20] and included in molecular breeding such as in *Exopalaemon carinicauda* [21] and *Ctenopharyngodon idella* [22] and other species. Many aquatic organisms have developed related SNPs through multi-omics sequencing, gene cloning and other technologies. However, there are still no reports on the SNP molecular markers of black porgy, which could be very useful for research on germplasm improvement, genetic diversity identification and growth performance optimization.

The purpose of this study was to perform genotyping of candidate SNPs in the transcriptome, conduct population verification, and analyze the association between different genotypes and growth traits of black porgy, so as to achieve the purpose of developing SNP molecular markers related to the growth trait of black porgy. To further carry out the genetic breeding and family identification of black porgy to provide an effective means.

## 2. Materials and Methods

### 2.1. Experimental Materials

The black porgy population used in this experiment for SNP locus genotyping and association analysis was hatched in the same batch, cultivated in the same pond in the breeding base of black porgy in Lvsi (Nantong, Jiangsu, China) and were fed twice a day. The diet used in the whole process of the experiment is made by Tongwei Co., Ltd. (Sichuan, China) and contains 42% fish meal protein. During the cultivation period, the depth of the pond was 2 meters high, and the density of the population was about 60 ind·M^2−^. The salinity of seawater was maintained at about 20–21‰, the average water temperature was 21 °C, the highest temperature was 33 °C and the lowest was 5 °C, which were all within the temperature range suitable for black porgy culture. After 120 days of rearing, 158 black porgy were randomly selected and their growth traits such as body weight (BW), full length (FL), body length (BL) and body height (BH) were measured, numbered and recorded. The caudal fins were cut and stored in absolute ethanol for DNA extraction. The number of the caudal fin corresponds to the growth data for each individual.

### 2.2. Experimental Method

#### 2.2.1. Extraction of Genomic DNA from Black Porgy

The extraction steps for black porgy DNA were carried out according to the instructions of the kit (Sangon Bio, Shanghai, China). The extracted DNA was tested for integrity, and concentration was estimated by 1% agarose gel electrophoresis and nucleic acid analyzer.

#### 2.2.2. Screening of Candidate SNPs

According to the transcriptome data results from our previous study [23], on the basis of differentially expressed genes, the whole genome of black porgy was used as a reference and GATK4 [24] software (Broad institute, Cambridge, MA, USA) was used to detect single nucleotide mutations. We also annotate and count the mutation position, type and function of the SNP sites. Transcriptome data have been uploaded to the NCBI database (BioProject ID: PRJNA686220) (accessed on 18 December 2020).

#### 2.2.3. Genotyping Primer Design

According to the position of the screened candidate SNP sites and the sequence of about 200 bp before and after the corresponding position in the whole genome of black porgy, genotyping tools and MassARRAY assay design software (Sequenom, San Diego, CA, USA) were used to design the PCR amplification primers and single base extension primer. The designed primers are in the range of 18–30 bp in length and the PCR product is about 80–200 bp. The Tm values of the primers were in the range of 55–65 °C and the annealing temperature was about 60 °C. The GC content of the primers was in the range of 40–70%. In addition, special care was taken to avoid the presence of primer dimers and non-specific amplification.

#### 2.2.4. PCR Amplification and Extension Reactions

On a 384-well plate, multiplex PCR technology was used for PCR amplification reactions. The PCR amplification reaction program was as follows: firstly 94 °C pre-denaturation for 3 min, then 94 °C denaturation for 30 s, 56 °C annealing for 25 s, and 72 °C extension for 30 s, a total of 40 cycles; finally, extend at 72 °C for 3 min.

After the PCR reaction, the PCR product was subjected to SAP (shrimp alkaline phosphatase) treatment to remove dNTPs in the reaction. The formula of the SAP reaction solution is shown in the following table (Table 1). The “SAP” reaction program was 37 °C for 40 min and 85 °C for 5 min. Finally, the iPlex reaction solution (Table 2) was added to the PCR product after alkaline phosphatase treatment, it was placed in the PCR machine, the program was run and the extension reaction carried out. The extension conditions were as follows: pre-denaturation at 94 °C for 30 s, denaturation at 94 °C for 5 s, annealing at 52 °C for 5 s and extension at 80 °C for 5 s; a total of 40 cycles. Then extension at 72 °C for 3 min was performed.

#### 2.2.5. SNP Locus Genotyping

Sangon Bioengineering Co., Ltd. (Shanghai, China) was entrusted to perform genotyping of candidate SNP loci by MALDI-TOF MS. MALDI-TOF MS is a representative of medium-throughput SNP genotyping technology. It has the characteristics of high sensitivity, rapid analysis, easy to analyze spectra and small sample consumption. It is a powerful analytical tool in the field of molecular biology [25,26]. MALDI-TOF MS is based on the “primer extension method”, which extends a base on the SNP site to be detected, and then is excited by instantaneous nanosecond (10^−9^ s) intense laser light and separated according to mass-to-charge ratio in a non-electric field drift region. Different genotypes can be distinguished according to the quality of the extended bases, which fly in the vacuum tube and arrive at the detector at different times [27].

#### 2.2.6. Genetic Polymorphism Analysis

Gene frequency, genotype frequency, homozygosity (*Ho*), heterozygosity (*He*), effective number of alleles (*Ne*) and polymorphism information content (*PIC*) parameters were calculated using Genepop32 software 4.2.2. Among them, the parameter of *Ne* is the reciprocal of gene homozygosity, which is an index to measure the genetic variation of the population. The more even the distribution of alleles in the population, the closer the *Ne* is to the actual detected allele frequencies. At the same time, the theoretical frequency of each genotype is calculated according to the allele frequency, so as to carry out the chi-squared test of the population Hardy–Weinberg equilibrium state [28,29].

#### 2.2.7. Correlation and Principal Component Analysis of Growth Traits

SPSS Statistics software 22.0 was used to analyze the correlation between BW, FL, BL and BH. It was also used for principal component analysis to find the first principal component trait from these four growth traits.

#### 2.2.8. Correlation and Amino Acid Sequence Analysis

Based on the general linear model, one-way ANOVA using SPSS Statistics software 22.0 was used to analyze the correlation between different genotypes of SNP and the growth traits of black porgy individuals. Using the different genotypes of SNPs as factors and using the traits of BW, FL, BL and BH of black porgy as dependent variables, the differences in growth traits between different genotypes were tested by counting the number of different genotypes mapped to four growth characters in each SNP. The least significant difference (LSD) method was used for post-hoc multiple comparisons [30].

DNAMAN software 9.0 was used to predict the amino acid sequence of the SNPs developed from this research.

## 3. Results and Analysis

### 3.1. Transcriptome Candidate SNPs Screening

According to our previous study [23], the experimental results of transcriptome sequencing of mixed tissues of black porgy with different growth rates showed that 3104 genes were distinguished to be significantly differentially expressed (FDR < 0.05 and |log2FC| > 1), of which, 1129 were up-regulated and 1975 were down-regulated genes. Based on the DEGs in the transcriptome and the whole genome sequence of the black porgy as a reference, GATK software 4.0 was used to detect the mutation of SNPs and Indels. A total of 572,598 SNP sites were detected. Data with base quality values lower than Q30 were removed and 109,719 high-accuracy black porgy SNP sites were screened. Among them, transition type SNPs accounted for 62.29% of the total sites, including G→A, C→T, A→G and T→C; transversion type SNPs accounted for 37.71% of the total sites, including C→G, G→ C, T→G, A→C, C→A, G→T, A→T and T→A (Figure 1). Based on the following filtering conditions: (1) the quality of the mutated base Q ≥ 900, (2) the difference between the number of reference reads and the number of mutated reads is ten times or more, (3) the mutated base belongs to a sense mutation, a total of 61 SNPs were selected for genotyping in this experiment. The primer information is shown in Appendix A.

### 3.2. Time-of-Flight Mass Spectrometry

The 61 candidate SNPs were genotyped using MALDI-TOF MS and the verification population sample was the genomic DNA of 158 randomly selected 4-month-old black porgy from the same batch of hatchings. The results show that a total of 60 SNPs can be successfully genotyped and the genotyping success rate is 98.36%. Among them were scaffold2-3426830, scaffold3-16653046, scaffold5-12042944, scaffold19-709400 and scaffold305-32581; these 5 SNPs have only two genotypes. The genotyping results of some SNPs are shown in the following figures: Figure 2, Figure 3 and Figure 4 included.

### 3.3. Polymorphism and Genetic Parameter Analysis of SNPs

Genepop32 software 4.2.2 was used to analyze the genetic parameters of sixty SNPs that were typed successfully, including gene frequency, genotype frequency, *Ho*, *He*, *Ne* and *PIC*, and to carry out the chi-squared test (*χ^2^*) of the population Hardy–Weinberg equilibrium state. As the results show in Appendix A, the *PIC* values of the SNPs ranged from 0.019 to 0.993. Among them, fifteen SNPs were low polymorphism sites (*PIC* < 0.25), such as scaffold10-3011393 and scaffold102-1223941. Twelve of them belong to moderate polymorphism SNPs (0.25 ≤ *PIC* < 0.5), such as scaffold12-12716321 and scaffold141-357706. Thirty-three SNPs were highly polymorphic sites (*PIC* ≥ 0.5), such as scaffold10-3011393 and scaffold11-8650275. The chi-squared test showed that 48 SNPs were in Hardy–Weinberg equilibrium (*p* > 0.05, *χ*^2^ < 5.99), including scaffold10-3011393, scaffold12-12716321 and scaffold13-4787950. Eight SNPs, including scaffold102-1223941 and scaffold13-4805623, deviated significantly from Hardy–Weinberg equilibrium (*p* < 0.01, *χ*^2^ > 9.21).

To summarize the results of the *PIC* and the Hardy–Weinberg equilibrium test, a total of thirty-seven moderately polymorphic or highly polymorphic SNPs that conform to the Hardy–Weinberg equilibrium (PIC ≥ 0.25 and *χ*^2^ < 5.99) were found, including scaffold10- 3011393, scaffold11-8650275 and scaffold12-12716321.

### 3.4. Correlation and Principal Component Analysis of Growth Traits

In order to verify the correlation between the four growth shapes and extract their principal component characters, we conducted correlation analysis and principal component analysis through SPSS Statistics 22. As the results show in Table 3, through the correlation analysis of 4 growth traits, the BW and BH were significantly correlated (*p* < 0.05). BW and FL, BW and BL, FL and BL, FL and BH, and BL and BH all show extremely significant correlations (*p* < 0.01). This result shows the significant correlation between the four growth traits, which meets the requirements of the experimental analysis and the conditions for further analysis. The dimensionality reduction and factor analysis were performed on the growth trait data of 158 black porgies. The results show that the BW accounted for 73.996% of the variance and the FL accounted for 24.977% of the variance, whereas the BL and BH accounted for less than 1%. According to the principle of principal component extraction, a factor with an eigenvalue greater than 1 is selected as the principal component. The initial eigenvalue of BW was 2.96, whereas the initial eigenvalues of FL, BL and BH were all less than 1. So, the BW was the first principal component of growth traits, with a cumulative variance of 73.996% (Table 4).

### 3.5. Correlation Analysis of SNPs Markers and Growth Traits

Correlation analysis was carried out between the different genotypes of SNPs and the four growth traits, BW, FL, BL and BH. The SNPs involved in the calculation were thirty-seven moderately or highly polymorphic loci that passed the Hardy–Weinberg equilibrium test. The results show that the different genotypes of eight SNPs had significant differences in one or more growth traits (Table 5). Among them, scaffold66-172323 only had a significant difference in BH and the average BW for the AG genotype of scaffold66-172323 was significantly higher than that for the AA genotype (*p* < 0.05). The average FL and BL for the AA genotype of scaffold168-178120 were significantly higher than for the GG genotype (*p* < 0.05). The TT genotype and CC genotype of scaffold12-12716321 and the AA genotype and CC genotype of scaffold13-4787950 had significant differences in BW, FL and BL (*p* < 0.05). The mean values for the TT genotype and CT genotype in scaffold290-11890 were significantly higher than those for the CC genotype on the four growth traits (*p* < 0.05). The mean values for the TT genotype of scaffold2-13687576 in all four growth traits were significantly higher than those for the AA genotype (*p* < 0.05). The calculated results for scaffold184-115262 and scaffold59-960994, which had less than two samples of partial genotypes, were not enough to explain the correlation between different genotypes and growth traits.

However, BW was the first principal component trait, under the conditions that both the requirements of the PIC and the Hardy–Weinberg equilibrium are satisfied, only four SNPs had significant differences between different genotypes for BW (scaffold12-12716321, scaffold13-4787950, scaffold2-13687576 and scaffold290-11890). Combined with the results of the association analysis of other growth traits, scaffold12-12716321 and scaffold13-4787950 had significantly higher averages for FL and BL than those of the CC genotype. However, there was no significant difference between the different genotypes of these two SNPs on the growth trait of BH. Scaffold2-13687576 and scaffold290-11890 had significant differences in FL, BL and BH. The averages of the individuals with the dominant genotype were higher than that of the individuals with the non-dominant genotype.

### 3.6. Amino acid Sequence Analysis

DNAMAN software 9.0 has been used to predict the amino acid sequences of the four SNPs developed from this research. The results show that scaffold13-4787950 was located at the 364th base from the 5’ end of the cDNA chain of cytochrome P450 3A (CYP3A40) and the base mutation type was A/C. Before mutation, the codon corresponding to this site was AAC and the translated amino acid was aspartic acid. The codon corresponding to this site was CAC after mutation and the translated amino acid becomes histidine (Figure 5). scaffold2-13687576 was located at the 6070th base of the cDNA strand from the 5’ end of the complement component 4 gene (C4). Before the mutation, the corresponding codon of this site was AAC. After the mutation, the corresponding codon was UAC. The resulting amino acid was changed from aspartic acid to tyrosine (Figure 6). Scaffold290-11890 was located at the 757th base on the fibrinogen beta chain (FGB). Before the mutation, the corresponding codon of this site was UUC, after the mutation, the corresponding codon was CUC. The translated amino acid was changed from phenylalanine was to leucine (Figure 7).

## 4. Discussion

### 4.1. Development of SNPs Markers by Transcriptome Sequencing

The RNA-Seq technique is a kind of molecular biology technology, which is deeply used in studying the signaling pathways and the molecular mechanisms of trait regulation of individual organisms at the transcriptional level. Moreover, with the continuous reduction of the cost of high-throughput sequencing, RNA-Seq technology has gradually played an important role in the screening of candidate functional genes and SNPs. However, the SNPs detected in transcriptome data often have a large number of false positive markers. Former studies have shown that in sequencing data with the whole genome sequence as a reference, although there are a large number of false positive SNPs that have been detected by the GATK software 4.0in the sequencing data, most of them can be removed by depth filtering [31]. So, we can improve the accuracy of results of genotyping and correlation analysis. In this research, high-throughput sequencing was performed on the total RNA from mixed tissues of brain, liver and muscle of 4-month-old black porgy hatched in the same batch with different growth rates. As the result, a total of 3038 M data were obtained and 18,180 genes were assembled. The average length of each gene was about 1910 bp. Through depth filtering, a total of 109,719 SNPs were obtained, with an average of one base mutation site per 316 bp. The density of SNPs in black porgy was significantly higher than that of *M. salmoides* [32] (1/2kb) and *C. idella* [33] (1/3.73 kb). It was close to *Megalobrama amblycephala* [4] (1/302bp), *Cheilinus undulatus* [34] (1/490bp) and *Litopenaeus vannamei* [35] (1/486bp) but significantly lower than that of *Schizothorax prenanti* [36] (1/85bp). The differences in the density of SNPs in different species may be affected by many factors such as species, sequencing depth and data screening criteria, but maybe the species factor is the most important one.

Compared with RT-qPCR, SNaPshot, HRM and other genotyping techniques, MALDI-TOF MS has the characteristics of high throughput, low cost, and low requirements for sample quality due to the simultaneous detection of multiple samples and multiple sites. In addition, it less time-consuming, highly sensitivity and has stronger compatibility [37]. In this experiment, the genotyping success rate was as high as 98.36%. This result demonstrates once again that depth filtering is an effective way to substantially pick out false-positive SNPs. Compared with related studies, 45 candidate SNPs from *Scophthalmus maximus* were genotyped using the HRM technique and 21 of them were successfully genotyped, with a success rate of 46.7% [38]. Researchers have successfully typed out 35 from 50 SNPs in *M. salmoides* using the SNaPshot method, with a success rate of 70% [28]. Sanger sequencing has also been used to type and verify 11 candidate SNPs in *Larimichthys crocea*, of which, 9 were successfully typed, a success rate of 81.8% [39]. The high success rate of genotyping results obtained in this study may be related to factors such as different genotyping techniques and sample treatment processing.

### 4.2. Genetic Parameter Analysis of SNPs in the Black Porgy Population

*Ne*, *He* and *PIC* are all important genetic parameters in population genetics and variation analysis. *He* is the frequency of heterozygotes in the population at the tested locus, which can reflect the degree of genetic variation in the population. A decrease of *He* indicates that the degree of genetic variation of the population has become lower, so as to judge that the genetic diversity of the population is becoming lower [40]. *PIC* is the reciprocal of genetic homozygosity, which can reflect the degree of mutual influence between alleles and is used as one of the evaluation parameters of population genetic diversity. According to the PIC’s judgment criteria [41], when the *PIC* ranges from 0.25 to 0.5, it is classified as moderately polymorphic, and when the *PIC* is higher than 0.5 or lower than 0.25, it is classified to be highly polymorphic or low polymorphic, respectively. In this study, the *PIC* values of our sixty SNPs ranged from 0.019 to 0.993. Among them, fifteen were low polymorphism sites (*PIC* < 0.25). Twelve of them belong to moderate polymorphism SNPs (0.25 ≤ *PIC* < 0.5) and thirty-three SNPs were highly polymorphic sites (*PIC* ≥ 0.5). The *He* of the black porgy population from those thirty-three highly polymorphic SNPs ranged from 0.096 to 0.499, with an average of 0.366. The average *He* of the eight SNP markers in the myosin heavy chain gene of *M. salmoides* was 0.41 [32]. The *He* of the four SNPs in the AMY gene of *S. chuatsi* was 0.45 [42]. The above-related results were similar to the results of this experiment, which proves that genetic diversity of the black porgy population used in this study is high.

### 4.3. Principal Component Analysis of Growth Traits of Black Porgy

In this study, the body size of black porgy when it grew to 4 months old was used as the standard for evaluating the growth rate. The results of principal component analysis showed that BW was the first principal component, with a cumulative contribution rate of 74%. The results of correlation analysis among the four growth traits showed that, as the first principal component, BW had a highly significant correlation with FL and BL (*p* < 0.01) and a significant correlation with BH (0.952, *p* < 0.05). FL, BL and BH were all highly significantly correlated with each other (*p* < 0.01). Compared with the analysis results of related studies on other species, the total length, body length and body height of the 12-month-old *Takifugu rubripes* showed a very significant correlation with body weight (*p* < 0.01) [43]. The full length, body length and body height of *S. prenanti* were all significantly correlated with the body weight, which was the first principal component trait, and the cumulative contribution rate was as high as 93.42% (*p* < 0.01) [44]. All these studies were consistent with the results of this study. Therefore, BW can be used as the main evaluation index when the association analysis of SNPs and growth traits is carried out.

### 4.4. Association Analysis of SNPs and Growth Traits of Black Porgy

According to the results of association analysis, among the thirty-seven moderately or highly polymorphic SNPs that passed the Hardy–Weinberg equilibrium test, there were eight different SNP genotypes that were correlated with one or more traits of black porgy. However, BW was the first principal component trait, under the condition that both the requirements of the *PIC* and the Hardy–Weinberg equilibrium are satisfied, only four SNPs have significant differences on BW between different genotypes. On the growth trait of BW, the dominant genotype of scaffold12-12716321, scaffold13-4787950 and scaffold2-13687576 were TT, AA and TT, respectively. CT and TT were the dominant genotypes of scaffold290-11890. According to the growth-related research in other species, a marker-assisted BLUP method was used to verify a SNP marker that was significantly correlated with body weight of turbot [11].

In addition, combined with the results of the association analysis of other growth traits, scaffold12-12716321, scaffold13-4787950, scaffold2-13687576 and scaffold290-11890 can be preliminarily confirmed as SNP molecular markers related to growth traits of black porgy, which can be used for the molecular breeding of black porgy to select individuals with a high growth rate.

### 4.5. The Effect of Base Changes on Amino Acid Sequence

The sequence of bases in DNA in each organism has an important impact on the amino acid sequence it translates. The mutation of bases may lead to changes in codons, thereby changing the types of amino acids and ultimately affecting protein structure. DNA sequencing of *I. punctatus* showed that 13 male-specific SNPs were identified in the coding region of the zbtb38 gene. Six of these substitutions resulted in changes in the amino acid coding of zbtb38 on the X and Y chromosomes [17]. Four SNPs were developed from this research. Among them, scaffold13-4787950 was located at the 364th base from the 5’ end of the cDNA chain of cytochrome P450 3A (CYP3A40) and the base mutation type was A/C. The amino acid type before and after mutation at this site changed from aspartic acid to histidine. Cytochrome P450 is mainly distributed in the liver and involved in the metabolic transformation of many exogenous and endogenous substances. CYP3A is the most abundant of the cytochrome P450 family and plays an important role in the detoxification of exogenous toxins and the metabolism of endogenous hormones in the liver [45]. Studies have shown that CYP3A40 is not only expressed early in development but remains expressed throughout adulthood. CYP3A is highly conserved in teleost fish and the gene sequence has high similarity in medaka, rainbow trout, zebrafish, grass carp and other species [46,47,48,49]. In this study, the base at scaffold13-4787950 was mutated, resulting in the corresponding amino acid changing from aspartic acid to histidine. According to the transcriptome sequencing data, the expression of the CYP3A40 gene was down-regulated in the slow-growing individual group. Therefore, we speculate that the base change at scaffold13-4787950 may affect the expression of this gene, thereby affecting the growth of black porgy. The specific mechanism and function need to be further studied.

scaffold2-13687576 was located at the 6070th base of the cDNA strand from the 5’ end of the complement component 4 gene (C4). Before and after the mutation, the resulting amino acid was changed from aspartic acid to tyrosine. Scaffold290-11890 was located at the 757th base on the fibrinogen beta chain (FGB). Before and after the mutation, the translated amino acid was changed from phenylalanine to leucine. Both C4 and FGB play important roles in the fish immune system. Fibrinogen is composed of three pairs of different polypeptide chains, α, β and γ. It is a glycoprotein synthesized and secreted by hepatocytes and is involved in the processes of coagulation and hemostasis. A large number of research results have shown that the damage, infection and immunity of organisms are related to extravascular fibrinogen. The results of research showed that the FGA gene is expressed in the liver tissue of *Salmo salar* after infection by A. salmonicida, which demonstrating that the FGA may play an important role in the innate immune response to bacterial invasion in Atlantic salmon [50]. The FGB of the large yellow croaker has homology with the FGB sequences of other species such as grouper and flounder through multiple sequence alignment. The large yellow croaker FGB was predicted to have biological functions similar to human FGB using a molecular model [51]. In this research, transcriptome sequencing results showed that the expression of the FGB gene was down-regulated in the slow-growing group and the base mutation at the scaffold290-11890 position changed its amino acid sequence. Therefore, we infer that due to the change in amino acid, the structure of the FGB protein may be changed, which will affect the immune function of black porgy to a certain extent, thereby affecting the growth rate.

## 5. Conclusions

Four SNP molecular markers related to the growth traits of black porgy were identified in this study, which means that we can select and breed fine varieties of black porgy from the molecular biology level. We overcome the limitations of high cost, low efficiency and poor effect caused by artificial breeding. This is conducive to improving the accuracy of selection and can identify individuals with excellent traits at an early stage, thereby shortening the breeding cycle and speeding up the breeding process.

## Figures and Tables

**Figure 1 genes-13-01992-f001:**
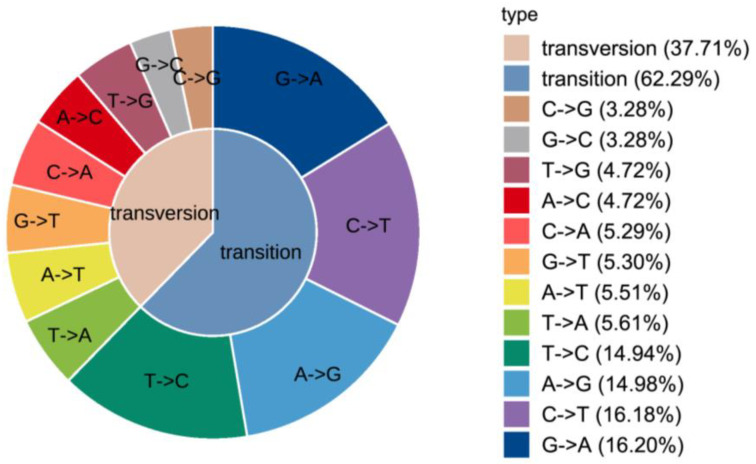
Statistics of mutation types in black porgy SNPs.

**Figure 2 genes-13-01992-f002:**
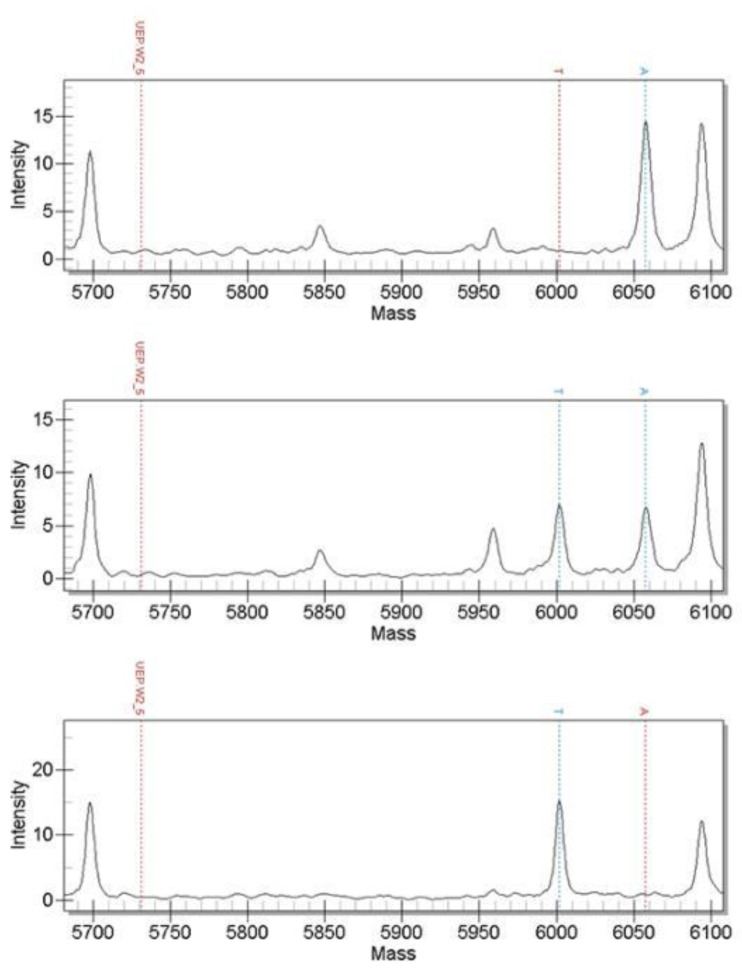
Mass spectrometry map of scaffold2-13687576 marker genotypes (genotypes are AA, AT and TT from top to bottom).

**Figure 3 genes-13-01992-f003:**
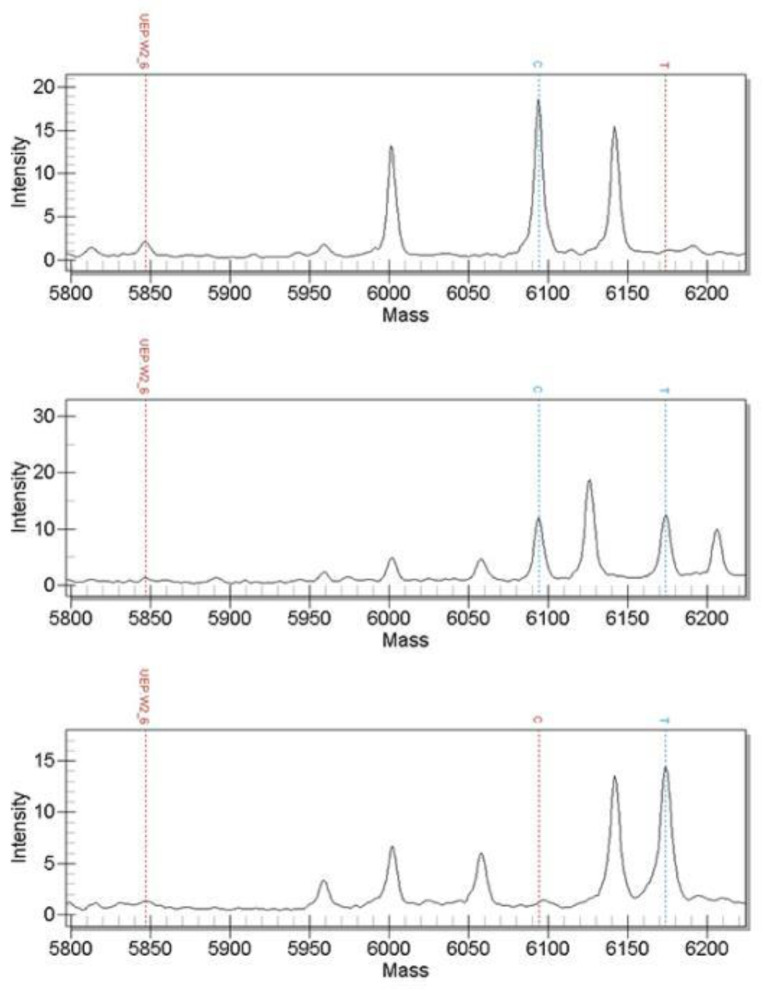
Mass spectrometry detection map of scaffold12-12716321 marker genotype (genotypes from top to bottom are CC, CT and TT.

**Figure 4 genes-13-01992-f004:**
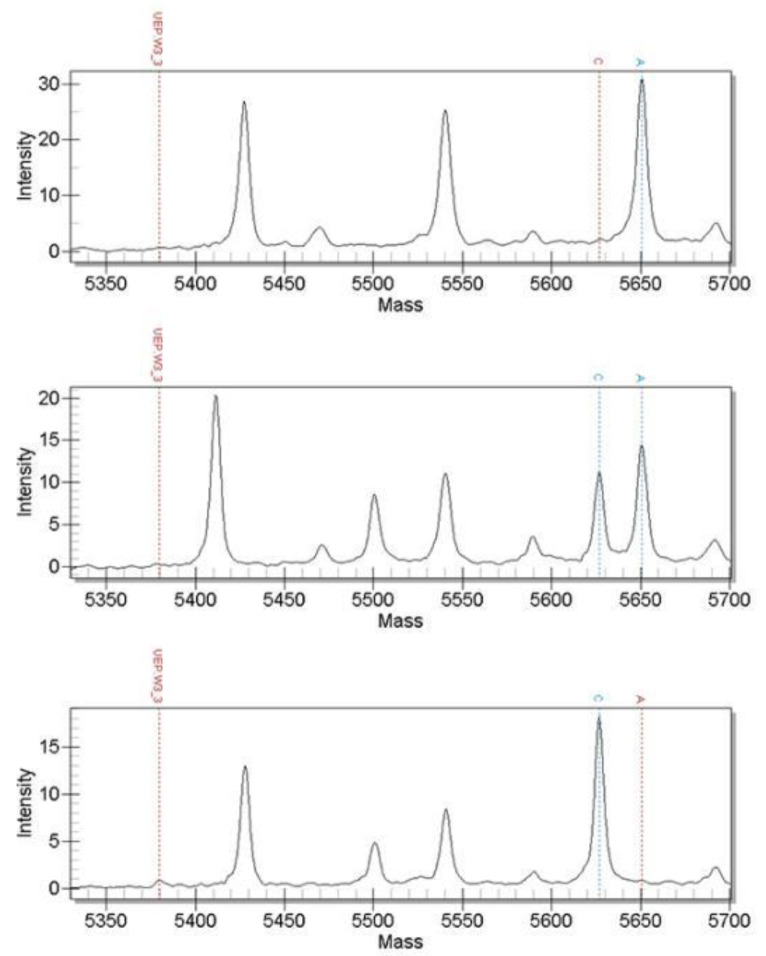
Mass spectrometry detection map of scaffold13-4787950 marker genotype (genotypes from top to bottom are AA, AC and CC).

**Figure 5 genes-13-01992-f005:**
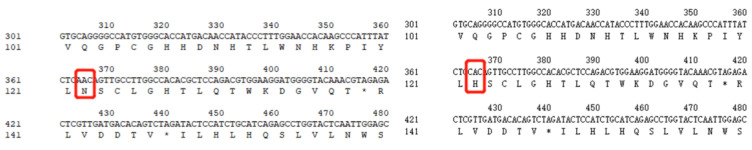
Changes in amino acid species with the base mutation before and after at scaffold13-4787950. N is aspartic acid and H is histidine □ as shown in the figure.

**Figure 6 genes-13-01992-f006:**
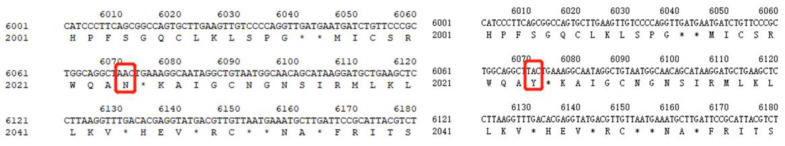
Changes in amino acid species with the base mutation before and after at the scaffold2-13687576. N is aspartic acid and Y is tyrosine □ as shown in the figure.

**Figure 7 genes-13-01992-f007:**
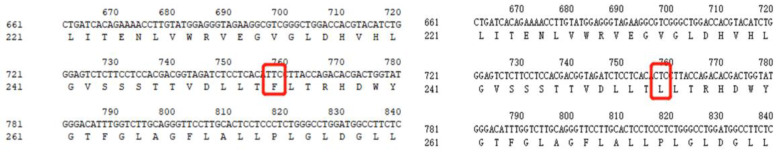
Changes in amino acid species with the base mutation before and after at scaffold290-11890. F is phenylalanine and L is leucine □ as shown in the figure.

**Table 1 genes-13-01992-t001:** The formula of SAP reaction solution.

Component	Concentration	Volume (1 rxn/μL)	Supplier
SAP Buffer	10×	0.17	ThermoFisher (Waltham, MA, USA)
SAP Enzyme	1.7 U/μl	0.3	ThermoFisher (Waltham, MA, USA)
Water	/	1.53	Sangon Bio (Shanghai, China)
PCR product	/	5	
Total volume		7	

**Table 2 genes-13-01992-t002:** The reaction system of primer extension.

Component	Concentration	Volume (1 rxn/μL)	Supplier
iPlex Buffer	10×	0.2	SEQUENOM (San Diego, CA, USA)
iPlex termination mix	10×	0.2	SEQUENOM (San Diego, CA, USA)
iPlex enzyme	2 U/μl	0.041	SEQUENOM (San Diego, CA, USA)
SAP Enzyme	1.7	0.3	ThermoFisher (Waltham, MA, USA)
Extension Primer	10 μm	0.804	Sangon Bio (Shanghai, China)
Water	/	0.755	Sangon Bio (Shanghai, China)
Purified PCR product	/	7	
Total volume		9	

**Table 3 genes-13-01992-t003:** Correlation analysis of growth traits of 4-month-old black porgy.

Traits	Body Weight (BW)	Full Length (FL)	Body Length (BL)	Body Height (BH)
BW	1			
FL	0.970 **	1		
BL	0.974 **	0.994 **	1	
BH	0.952 *	0.969 **	0.969 **	1

Note: * indicates significant correlation between phenotypic traits (*p* < 0.05), ** indicates extremely significant correlation between phenotypic shapes (*p* < 0.01).

**Table 4 genes-13-01992-t004:** Principal component analysis of growth traits of black porgy.

Component	Initial Component			Sum of Squares of Extracted Component		
	Total	Variance Ratio/%	Accumulation Variance Ratio/%	Total	Variance Ratio/%	Accumulation Variance Ratio/%
BW	2.960	73.996	73.996	2.960	73.996	73.996
FL	0.999	24.977	98.972			
BL	0.035	0.882	99.854			
BH	0.006	0.146	100.000			

**Table 5 genes-13-01992-t005:** Correlation analysis of different genotypes of SNPs and growth traits of 4-month-old black porgy.

SNPs	Type of Mutation	Genotypes	Number of Samples	BW/cm	FL/cm	BL/cm	BH/cm
scaffold12-12716321	T > C	CC	101	12.89 ± 8.25 ^b^	8.98 ± 1.88 ^b^	7.46 ± 1.58 ^b^	2.82 ± 0.70
		CT	51	14.15 ± 7.93 ^ab^	9.33 ± 1.87 ^ab^	7.74 ± 1.62 ^ab^	2.91 ± 0.67
		TT	6	19.58 ± 6.26 ^a^	10.45 ± 1.32 ^a^	8.80 ± 1.12 ^a^	3.27 ± 0.53
scaffold13-4787950	A > C	AA	29	16.13 ± 7.35 ^a^	9.72 ± 1.74 ^a^	8.10 ± 1.46 ^a^	3.04 ± 0.60
		AC	65	13.78 ± 9.14 ^ab^	9.18 ± 1.99 ^ab^	7.62 ± 1.69 ^ab^	2.88 ± 0.74
		CC	64	12.14 ± 7.19 ^b^	8.85 ± 1.76 ^b^	7.36 ± 1.51 ^b^	2.77 ± 0.64
scaffold168-178120	A > G	AA	28	15.67 ± 7.60	9.71 ± 1.79 ^a^	8.02 ± 1.49 ^a^	3.01 ± 0.62
		AG	76	13.74 ± 7.39	9.24 ± 1.76 ^ab^	7.68 ± 1.51 ^ab^	2.89 ± 0.66
		GG	54	12.19 ± 9.27	8.73 ± 2.00 ^b^	7.28 ± 1.71 ^b^	2.75 ± 0.74
scaffold184-115262	A > G	AA	1	4.17 ± 0.00 ^b^	6.40 ± 0.00 ^b^	5.20 ± 0.00 ^b^	1.90 ± 0.00 ^b^
		AG	14	13.35 ± 7.88 ^a^	9.09 ± 1.98 ^a^	7.51 ± 1.63 ^a^	2.86 ± 0.72 ^a^
		GG	143	13.63 ± 8.19 ^a^	9.17 ± 1.86 ^a^	7.63 ± 1.58 ^a^	2.87 ± 0.68 ^a^
scaffold2-13687576	A > T	AA	2	4.87 ± 1.18 ^b^	6.7 ± 0.81 ^b^	5.75 ± 0.52 ^b^	2.15 ± 0.17 ^b^
		AT	34	10.96 ± 6.82 ^ab^	8.63 ± 1.71 ^a^	7.09 ± 1.41 ^a^	2.67 ± 0.61 ^ab^
		TT	122	14.41 ± 8.35 ^a^	9.33 ± 1.88 ^a^	7.77 ± 1.60 ^a^	2.93 ± 0.69 ^a^
scaffold290-11890	T > C	CC	19	9.46 ± 4.34 ^b^	8.30 ± 1.29 ^b^	6.87 ± 1.07 ^b^	2.53 ± 0.47 ^b^
		CT	76	14.26 ± 8.77 ^a^	9.31 ± 1.97 ^a^	7.74 ± 1.69 ^a^	2.92 ± 0.72 ^a^
		TT	63	13.92 ± 7.98 ^a^	9.20 ± 1.86 ^a^	7.65 ± 1.56 ^a^	2.90 ± 0.67 ^a^
scaffold59-960994	A > G	AA	1	4.09 ± 0.00 ^b^	6.50 ± 0.00 ^b^	5.50 ± 0.00 ^b^	1.08 ± 0.00 ^b^
		AG	33	15.21 ± 10.30 ^a^	9.48 ± 2.13 ^a^	7.89 ± 1.81 ^a^	2.97 ± 0.75 ^a^
		GG	124	13.18 ± 7.46 ^a^	9.08 ± 1.79 ^a^	7.54 ± 1.52 ^a^	2.85 ± 0.65 ^a^
scaffold66-172323	A > G	AA	3	8.44 ± 6.04	8.03 ± 1.93	6.67 ± 1.46	2.19 ± 1.16 ^b^
		AG	41	14.89 ± 6.85	9.53 ± 1.61	7.90 ± 1.37	3.05 ± 0.58 ^a^
		GG	114	13.20 ± 8.58	9.04 ± 1.95	7.52 ± 1.66	2.81 ± 0.69 ^ab^

Note: The values in the table are composed of mean ± standard error. In the same column of values, the superscript with the same letter indicates that the difference between the two genotypes is not significant (*p* > 0.05) and different letters indicate significant difference (*p* < 0.05).

## Data Availability

The data that support the findings of this study are all generated by our experiments. The raw transcriptome data has been uploaded to the NCBI Sequence Read Archive (SRA). The BioProject ID and Submission ID are PRJNA686220 and SUB8756480. https://www.ncbi.nlm.nih.gov/bioproject/?term=+PRJNA686220 (accessed on 18 December 2020).

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
