# Peer review of "Development of Single Nucleotide Polymorphism and Association Analysis with Growth Traits for Black Porgy (Acanthopagrus schlegelii)"

_genes, 2022, doi:10.3390/genes13111992_

Round 1
Reviewer 1 Report
Abstract: I suggest including a sentence about Black porgy aquaculture and its importance.
Materials and methods:
I think that some information about diet used during the experiment as well as fish density should be included. In addition fish initial conditions are missing.
Experimental methods are not clear and should be described better.
Tables about PCR conditions should be deleted.
Please confirm that SPSS Statistics 22 was used for genetic population parameters estimation. In general other softwares are used for these estimations.
Results:
Tables 1 and Table 2 can be included as supplementary tables as well as Figures 5, 6 and 7.
Some results and methodology are described at the Discussion section. They should be shown at M and M and Results section.
Author Response
Ponit 1:Abstract: I suggest including a sentence about Black porgy aquaculture and its importance.
Response 1: The total production of black porgy is about 60,000 tones in one year in China, which is at the fifth position in all kinds of marine-cultured fishes just following the large yellow croak, sea bass, grouper, flounder. The black porgy aquaculture has been formed complete industry chain from its breeding, aquaculture, processing and sales, which play an important role in the marine fish culture.
Ponit 2:Materials and methods:
I think that some information about diet used during the experiment as well as fish density should be included. In addition fish initial conditions are missing.
Experimental methods are not clear and should be described better.
Response 2: The diet used in the whole process of the experiment is made by Tongwei Co., LTD and contains the fish meal protein of 42%. These contents have been supplemented in section 2.1.
Ponit 3:Tables about PCR conditions should be deleted.
Response 3: The table has been deleted.
Ponit 4:Please confirm that SPSS Statistics 22 was used for genetic population parameters estimation. In general other softwares are used for these estimations.
Response 4: The software Genepop32 was used for genetic population parameters estimation.
Ponit 5:Results:
Tables 1 and Table 2 can be included as supplementary tables as well as Figures 5, 6 and 7.
Response 5: They have been adjusted to the supplementary part.
Ponit 6:Some results and methodology are described at the Discussion section. They should be shown at M and M and Results section.
Response 6: We re-describe some results in the discussion section in order to compare them with other related research.so we think it does not repeat and does not need to move to the M and M section.

Reviewer 2 Report
Section 2.2.2.2 Authors need to be clearer in the results of their previous studies or cite them adequately.
The analysis of genetic polymorphisms as well as correlation and principal component analysis for growth traits should be better described.
Section 2.2.8 Clarify how correlations between markers and dependent variables were analyzed through ANOVA.
Table 4 is too large, it could go to an appendix of the publication.
The results and discussion are adequate. Some of the conclusions are written as results, they should be modified.
Author Response
Point 1: Section 2.2.2.2 Authors need to be clearer in the results of their previous studies or cite them adequately.
Response1: We mainly describe the experimental methods in section 2.2.2, and we gave a general description of the previous research results in section 3.1, which mainly describing the DEGs obtained from transcriptome sequencing data to explain the source of candidate SNPs in subsequent experiments.
Point 2: The analysis of genetic polymorphisms as well as correlation and principal component analysis for growth traits should be better described.
Response2: We have supplemented the description of genetic polymorphism analysis, correlation and principal component analysis of growth traits in section 3.4.
Point 3: Section 2.2.8 Clarify how correlations between markers and dependent variables were analyzed through ANOVA.
Response 3: The differences of growth characters among different genotypes were tested by counting the number of different genotypes mapped to four growth characters in each SNP. This part has been added to section 2.2.8.
Point 4: Table 4 is too large, it could go to an appendix of the publication.
Response 4: No problem.
Point 5: The results and discussion are adequate. Some of the conclusions are written as results, they should be modified.
Response 5: Thanks for your recognition! Some of the conclusions as results have been modified, and section 3.6 has been added to the results section.
